# Stylized Dialogue Generation with Feature-Guided Knowledge Augmentation

**Jinpeng Li[1][*], Zekai Zhang[1][*], Xiuying Chen[2], Dongyan Zhao[1,3,4][†], Rui Yan[5][†]**

[1]Wangxuan Institute of Computer Technology, Peking University
[2]King Abdullah University of Science and Technology
[3]State Key Laboratory of Media Convergence Production Technology and Systems
[4]Institute for Artificial Intelligence, Peking University
[5]Gaoling School of Artifical Intelligence, Renmin University of China

{lijinpeng,justinzzk}@stu.pku.edu.cn, xiuying.chen@kaust.edu.sa,
zhaody@pku.edu.cn, ruiyan@ruc.edu.cn

## Abstract

Stylized dialogue generation systems aim to produce coherent and context-aware dialogues while effectively emulating the desired style. Generating stylized dialogue is valuable yet challenging due to the scarce parallel data. Existing methods often synthesize pseudo data through back translation, yet suffer from noisy and context-agnostic style signals caused by insufficient guidance on target style features. To address this, we propose the knowledge-augmented stylized dialogue generation model, which includes a feature-guided style knowledge selection module that utilizes context and response features. Specifically, we retrieve dialogue-related style sentences from style corpus to explicitly provide clear style signals. We design a feature-guided selection module with response-related contrastive learning and style responsiveness Kullback-Leibler losses to enhance generation at both semantic and stylized levels. Our approach demonstrates satisfactory performance on two public stylized dialogue benchmarks in both automatic and human evaluations. We have released our code and datasets. [1]

## 1 Introduction

With the increasing demand for meaningful human-computer interaction, the development of advanced dialogue systems has become crucial across various domains. Stylized dialogue generation focuses on producing human-like conversations by generating stylized and coherent responses. To achieve this, significant efforts have been dedicated to building response-matching models that leverage both dialogue content and style sentences, resulting in more engaging and diverse responses (Gao et al., 2019b; Zheng et al., 2021; Li et al., 2021; Sun et al., 2022).

---

[*]Equal contribution.
[†]Corresponding author: Dongyan Zhao and Rui Yan.
[1]https://github.com/ZekaiGalaxy/KASDG

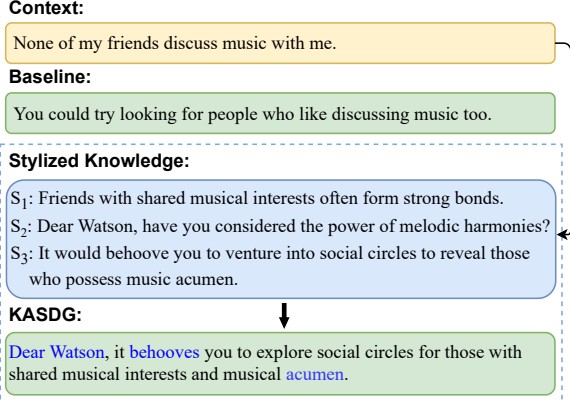

Figure 1: Example of stylized dialogue generation enhanced by knowledge from semantic and stylistic level.

However, stylized dialogue generation encounters a challenge due to the absence of parallel training data between context and desired stylized responses, which renders supervised methods ineffective. One natural solution to this problem is to enhance the connection between different hidden space vectors. For instance, Gao et al. (2019b) propose a method that bridges conversation modeling and non-parallel style transfer by sharing a structured latent space. Nevertheless, this approach is limited by the quality of the available data, and the weak supervision signal leads to unsatisfactory performance in practice (Zheng et al., 2021). Another direction is to establish pseudo pairs between contexts and style-specific responses using unsupervised or semi-supervised learning by back translation, which is proposed for translating a sentence from the target language back to the source language, allowing the model to learn from the differences and improve translation quality (Sennrich et al., 2016). Su et al. (2020) propose a diversifying dialogue model based on iterative back translation. Zheng et al. (2021) employ a style rout-

ing approach with a joint training process, where a backward model with style embeddings is used to generate pseudo pairs and train the stylized model. Li et al. (2021) bridge the forward and backward models through a text transfer dataset to construct high-quality pseudo-contexts. Despite the progress made by these approaches, the pseudo data constructed using the back translation method exhibit low diversity and these methods suffer from noisy and context-agnostic style signals (Li et al., 2021). This is because the style knowledge is not explicitly provided for each response. Additionally, there is a lack of alignment between relevant stylistic words and responses, significantly reducing the incorporation of style features in the generated responses.

Motivated by knowledge-grounded dialogue generation (Liu et al., 2021), we introduce a knowledge-augmented stylized dialogue generation (KASDG) model, as illustrated in Figure 1. Our model utilizes style corpus from an external knowledge base perspective, and provides style information for response generation in an explicit way. Specifically, we employ a style sentence retrieval process, which retrieves the most relevant style sentences based on similarity metrics to guide the style of response. In order to filter out noisy information and provide dialogue-related style signals, we refine the retrieved style feature using a feature-guided selection module, wherein the retrieved style feature is filtered under the guidance of context. Moreover, to reduce the gap between context-related selection and response-related selection and obtain more useful filtered style features for response generation, we propose response-related contrastive learning and style responsiveness Kullback-Leibler loss. The context hidden and filtered style features are integrated into the content-aware attention and style-aware attention in decoder, producing the final dialogue response.

We conduct extensive experiments on two benchmarks with four distinct styles to verify the effectiveness of our proposed model. The experimental results demonstrate that our approach generates more informative and accurate responses that align with the given stylized knowledge. We also discuss the advantages of our model compared with large language model in Appendix A.5. In summary, our contributions can be summarized as follows:

- We propose a knowledge-augmented stylized dialogue generation model with a feature-guided selection in an unsupervised manner.

- We design a feature-guided selection module to filter style features and bridge the gap between prior and posterior in style sentence selection.

- Experiments conducted on two datasets with four styles show that our proposed method outperforms all baselines with limited training data.

## 2 Related Work

**Stylized Dialogue Generation** aims to generate coherent dialogues with specific styles. The lack of supervised data leads to difficulties in style and semantic alignment, which affects the performance of the model, and the collection of parallel corpus is time-consuming and laborious. So it is critical to utilize unpaired style corpus to generate stylized responses. Previous research has explored style information in implicit ways. SFusion (Gao et al., 2019b) constructs structured latent space that aligns style examples into the neighborhood of response, providing clearer signals. StyleDGPT (Yang et al., 2020) leverages both a style language model and style classifier to provide style signals. Additionally, SRJT (Zheng et al., 2021) and MPDL (Li et al., 2021) use back translation techniques to obtain pseudo-contexts. However, these methods suffer from noisy and context-agnostic style signals, as the style is not explicitly provided for each response. Our approach explicitly exploits style information from an external knowledge perspective and can provide clearer style signals for better generation via knowledge retrieval and filtering.

**Knowledge Selection** focuses on identifying relevant knowledge sources from a large corpus to improve downstream tasks, which can be divided into two categories: sequential selection and non-sequential selection. The former aims to select knowledge based on previously selected ones, to gradually build a coherent knowledge base (Kim et al., 2020; Zheng et al., 2020). The latter is identified based on its relevance to the task at hand (Qin et al., 2019; Ren et al., 2019; Gao et al., 2019a; Chen et al., 2022, 2023). This requires the model to have a high ability to understand natural language (Cheng et al., 2023a,b). Liu et al. (2021) propose a three-stage learning framework that includes a controller to dynamically select knowledge during the decoding phase. Meanwhile, Chen et al. (2020) attempt to bridge the gap between prior and posterior knowledge selection. Inspired by these works, we leverage a retrieval and selection framework to dynamically select needed style signals for

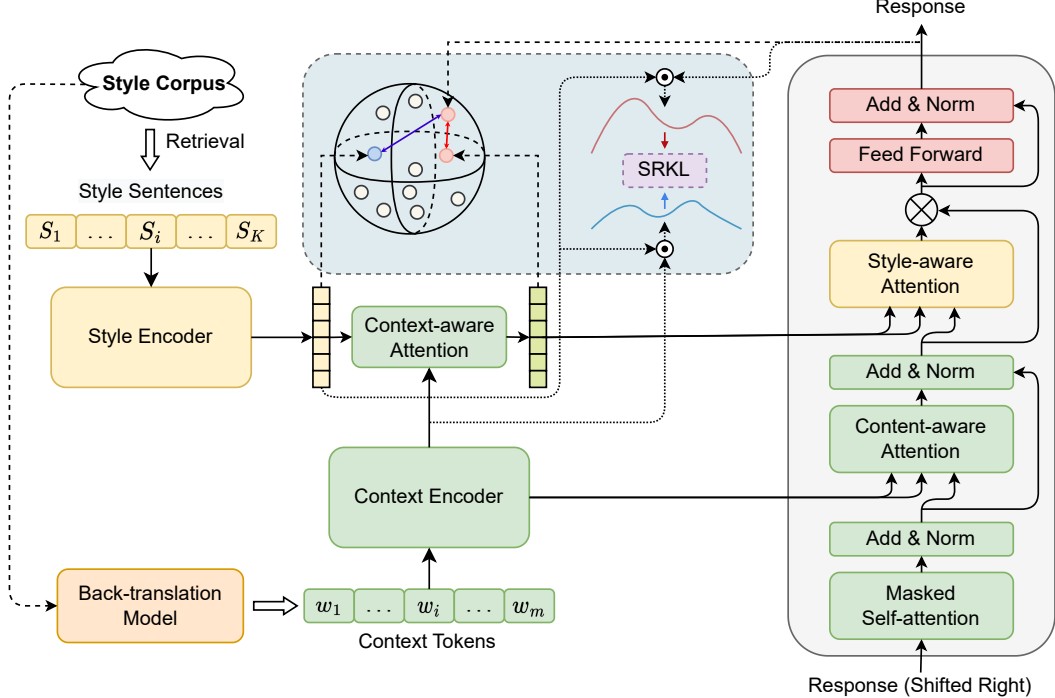

Figure 2: The framework of knowledge-augmented stylized dialogue generation (KASDG) model with a feature-guided selection module. The blue arrow maximizes distance between representations, while the red arrow minimizes distance between representations in contrastive learning.

guiding dialogue generation.

**Text Style Transfer** is to generate sentence that is both stylistic and preserves the original meaning, which focuses solely on style transfer without considering conversational coherence. With the scarcity of parallel corpus, recent approaches have focused on unsupervised text style transfer, which can be categorized into three main approaches: style disentanglement, prototype editing, and pseudo-parallel corpus construction (Jin et al., 2020). Style disentanglement approaches (Hu et al., 2017; Yi et al., 2020; Zhao et al., 2017; Fu et al., 2017; Li et al., 2020) learn a latent representation of input that separates the content and style information. This representation can be manipulated to perform style transfer while preserving the content. For instance, Fu et al. (2017) use adversarial networks to separate content and style representations, and change the style embedding to perform style transfer. Prototype editing approaches (Li et al., 2018; Madaan et al., 2020; Sudhakar et al., 2019; Wu et al., 2019; Jin et al., 2019a) identify style markers in the sentence and edit them to the target domain to perform style transfer. Pseudo-parallel corpus construction approaches (Jin et al., 2019b; Nikolov and Hahnloser, 2018; Zhang et al., 2018) generate pseudo-stylized

data using back-translation for further training. Our approach mainly follows the style disentanglement framework and utilizes back-translation method.

## 3   Methodology

### 3.1   Model Overview

The knowledge-augmented stylized dialogue generation model consists of two main components, the encoder-decoder framework and the feature-guided selection module, as shown in Figure 2. We employ separate style encoder and context encoder to encode style and content information, respectively. Inspired by the Liu et al. (2021), we add the style-aware attention and gate network to control the knowledge and context contributions. The model is trained on a dialogue dataset $\mathcal{D}_{dia} = \{(X_i, Y_i)\}_{i=1}^{M}$ and a style corpus $\mathcal{D}_{sty} = \{S_i\}_{i=1}^{N}$, where $X_i$ is the dialogue context with a relevant response $Y_i$ in style $s_0$, $S_i$ is the $i$-th sentence in the target style $s_1$. Given an input context $X_i = \{w_1, w_2, ..., w_m\}$ consisting of $m$ words, we aim to learn a model $f_\theta$ to generate stylized response $\hat{Y}_i$. Here, we propose to first retrieve context-related style knowledge $\mathcal{D}_s(X_i) \subset \mathcal{D}_{sty}$, and then combine them with context $X_i$ to generate stylized response $\hat{Y}_i = f_\theta(X_i, \mathcal{D}_s(X_i))$. The opti-

mization objectives of the forward model $f_\theta$ can be represented as follows:

$$\mathcal{L}_{dia}^{s_i} = -\log P(Y|X, \mathcal{D}_s(X); f_\theta), \quad (1)$$

where $i \in \{0, 1\}$. Besides, we have a backward model $f_\phi$ which can generate pseudo context according to the stylized sentence for maximizing the supervision. The difference with previous work is that our backward model is only optimized through $(Y_i, X_i)$. This operation is easier to converge, reducing the training overhead. Similarly, the optimization objectives of $f_\phi$ is as follows:

$$\mathcal{L}_{bt} = -\log P(X|Y; f_\phi). \quad (2)$$

## 3.2 Style Sentences Retrieval

In order to retrieve context-related style knowledge for the input, we apply an information retrieval module with sentence-BERT (Reimers and Gurevych, 2019), which use siamese and triplet network structures to derive semantically meaningful sentence embeddings. Specifically, we use sentence-BERT to encode context $X$ and the style corpus $\mathcal{D}_{sty} = \{S_i\}_{i=1}^N$ respectively and calculate the relevance score $Rel$ using cosine-similarity:

$$Rel_i = \text{sim}(\text{BERT}(X), \text{BERT}(S_i)), \quad (3)$$

where $\text{sim}(h_1, h_2) = \frac{h_1^T h_2}{||h_1|| \cdot ||h_2||}$. Then the most relevant sentences from $\mathcal{D}_{sty}$ are retrieved and sent to style encoder to obtain retrieved style hidden, serving as the style knowledge of the input. We also analyze the performance of dynamic retrieval and fixed retrieval in Appendix A.3.

## 3.3 Feature-Guided Selection Module

Inevitably, there exists noisy and irrelevant information in the retrieved style hidden. To address these redundancies and extract dialogue-related style information, we introduce the feature-guided selection module. Specifically, we integrate context information as guidance for our style feature selection process. For clarity, let the context hidden be $h_c \in R^{L_c \times d}$ and the original style hidden of the retrieved style sentences be $h_s \in R^{L_s \times d}$. The filtered style features obtained from our selection module are denoted as $h_f$, which is constructed utilizing the attention mechanism:

$$h_f = softmax(h_c h_s^T) h_s, \quad (4)$$

where $h_f \in R^{L_c \times d}$, $L_c$ and $L_s$ are the length of context and style sentences, respectively. Simply relying on contextual similarity to filter the retrieved style hidden can be ineffective, since the majority of the filtered style features tend to be context-related rather than response-related, reducing their effectiveness in response generation. To address this issue, we examine the relationship between context hidden $h_c$, response hidden $h_r$, retrieved style hidden $h_s$ and filtered style feature $h_f$. In Section 3.3.1, we propose response-related contrastive learning to make filtered style feature more response-related by exploring the interaction between $h_s$, $h_f$, and $h_r$. In Section 3.3.2, we introduce style responsiveness Kullback-Leibler loss to reduce the gap between context-guided selection and response-guided selection by exploring the interaction between $h_c$, $h_r$, and $h_f$.

### 3.3.1 Response-Related Contrastive Learning

We utilize contrastive learning (Gao et al., 2021) to make filtered style features more response-related. Following the intuition that filtered style features $h_f$ should be closer to response $h_r$ than the retrieved style hidden $h_s$, we design our response-related contrastive learning to supervise the distance relation. For a mini-batch of $N$ pairs of $(h_{s_i}, h_{r_i}, h_{f_i})$, $i$ denotes the $i$-th sample in the batch. We treat responses as the anchor, the corresponding filtered style features as positive examples, other filtered style features and all retrieved style hidden as negative examples. The response-related contrastive learning loss is defined as:

$$\mathcal{L}_{cl} = -\log \frac{e^{\text{sim}(h_{r_i}, h_{f_i})/\tau}}{\sum_{j=1}^N (e^{\text{sim}(h_{r_i}, h_{f_j})/\tau} + e^{\text{sim}(h_{r_i}, h_{s_j})/\tau})}, \quad (5)$$

where $\tau$ is the temperature of contrastive learning.

### 3.3.2 Style Responsiveness Kullback-Leibler

As we rely solely on context to filter retrieved style hidden, a potential gap emerges between context-guided and response-guided selection processes. To overcome this disparity, we introduce the style responsiveness Kullback-Leibler (SRKL) loss, designed to optimize filtered style features using both context and response. We formulate this problem as an effort to bridge the gaps between the prior and posterior distributions of style selection, utilizing cosine similarity as the importance metric. Specifically, context $h_c$ and response $h_r$ can assign different importance to different positions of $h_s = \{h_s^1, h_s^2, ..., h_s^{L_s}\}$. The context-guided prior distribution $p(s)$ and response-guided posterior dis-

tribution $q(s)$ are calculated as:

$$p(s_i) = \frac{\exp(h_s^{i\,T} \cdot h_c)}{\sum_{j=1}^{L_s} \exp(h_s^{j\,T} \cdot h_c)},$$

$$q(s_i) = \frac{\exp(h_s^{i\,T} \cdot h_r)}{\sum_{j=1}^{L_s} \exp(h_s^{j\,T} \cdot h_r)}, \tag{6}$$

The KL divergence loss is defined to be:

$$\mathcal{L}_{KL} = \sum_{i=1}^{L_s} p(s_i) log \frac{p(s_i)}{q(s_i)}, \tag{7}$$

According to Equation 6 we can obtain:

$$\log p(s_i) = h_s^{i\,T} \cdot h_c - \log(\sum_{j=1}^{L_s} \exp(h_s^{j\,T} \cdot h_c)),$$

$$\log q(s_i) = h_s^{i\,T} \cdot h_r - \log(\sum_{j=1}^{L_s} \exp(h_s^{j\,T} \cdot h_r)), \tag{8}$$

where $\sum_{i=1}^{L_s} p(s_i) = 1$. Hence, we can decompose KL loss into $\mathcal{L}_{KL} = \mathcal{L}_{dir} + \mathcal{L}_{resp}$ as follows:

$$\mathcal{L}_{dir} = (\sum_{i=1}^{L_s} p(s_i)h_s^i)^T(h_c - h_r),$$

$$\mathcal{L}_{resp} = \log \sum_{i=1}^{L_s} exp(h_s^{i\,T} \cdot h_r) - \log \sum_{i=1}^{L_s} exp(h_s^{i\,T} \cdot h_c), \tag{9}$$

where $\mathcal{L}_{dir}$ is the direction loss, and $\mathcal{L}_{resp}$ is the responsiveness loss.

**Direction Loss.** In $\mathcal{L}_{dir}$, the term $\sum_{i=1}^{L_s} p(s_i)h_s^i$ represents a content-guided reweighted style hidden, where the weight denotes importance. This concept aligns with the objective of our feature-guided selection module. To improve guidance signals, we replace the term with the filtered style feature $h_f$. Intuitively, the difference between the response and context, $h_r - h_c$, signifies the necessary style information for the generated response to become stylized. Thus, $h_f^T(h_c - h_r)$ can be deemed as a direction loss guiding the selected style features:

$$\mathcal{L}_{dir}^* = \max(0, -h_f^T(h_r - h_c)). \tag{10}$$

**Responsiveness Loss.** To evaluate the extent of information shared between the selected features and generated response, we define the responsiveness score $\mathcal{R}(x, y) = \max_i(x_i^T y)$, which examines the maximum information that hidden $x$ can provide

to a query $y$. $\mathcal{L}_{resp}$ can be approximated by:

$$\mathcal{L}_{resp} = \log \sum_{i=1}^{L_s} exp(h_s^{i\,T} \cdot h_r) - \log \sum_{i=1}^{L_s} exp(h_s^{i\,T} \cdot h_c)$$

$$\approx \max_i(h_s^{i\,T} \cdot h_r) - \max_i(h_s^{i\,T} \cdot h_c)$$

$$= \mathcal{R}(h_s, h_r) - \mathcal{R}(h_s, h_c). \tag{11}$$

From the perspective of responsiveness score, $\mathcal{L}_{resp}$ supervises that the retrieved style hidden shares more information with context than with response. However, this supervision offers limited assistance to the feature-guided selection module since our ultimate goal is to enhance the usefulness of $h_f$ for response. Therefore, we change the supervised object from $h_s$ to $h_f$. Moreover, with the intuition that the filtered style feature $h_f$ should share more information with response than context, so we reformulate $\mathcal{L}_{resp}$ as:

$$\mathcal{L}_{resp}^* = \max(0, \mathcal{R}(h_f, h_c) - \mathcal{R}(h_f, h_r)) \tag{12}$$

The SRKL loss can be formalized as a weighted sum of direction and responsiveness losses:

$$\mathcal{L}_{SRKL} = \lambda_{dir}\mathcal{L}_{dir}^* + \lambda_{resp}\mathcal{L}_{resp}^*, \tag{13}$$

where $\lambda_{dir}$ and $\lambda_{resp}$ are the weight scalars.

### 3.4 Training Objective

The training objective of the forward model can thus be defined as:

$$\mathcal{L}_{s_i} = \mathcal{L}_{dia}^{s_i} + \lambda_{cl}\mathcal{L}_{cl} + \lambda_{SRKL}\mathcal{L}_{SRKL}, \tag{14}$$

where $i \in \{0, 1\}$. The overall training objective can be expressed as:

$$\mathcal{L} = \lambda_{s_0}\mathcal{L}_{s_0} + \lambda_{s_1}\mathcal{L}_{s_1} + \lambda_{bt}\mathcal{L}_{bt}, \tag{15}$$

where $\lambda_{cl}, \lambda_{SRKL}, \lambda_{s_0}, \lambda_{s_1}$, and $\lambda_{bt}$ are weight scalars. In the experiments, we find that the back translation can be unstable, so we propose a warmup training strategy for the loss $\mathcal{L}_{s_1}$:

$$\lambda_{s_1} = \min(\max(\frac{T_c - T_f}{T_w - T_f}, 0), 1), \tag{16}$$

where $T_c, T_f, T_w$ represent the current training step, freeze step, and warm-up step respectively.

## 4 Experimental Setup

### 4.1 Datasets

We conduct our experiments on two benchmarks with four distinct styles: generating formal-like and informal-like responses (TCFC (Wu et al., 2020))

| Style | Model | BLEU-1 | BLEU-2 | Rouge-L | Dist-1 | Dist-2 | StyleIn. |
|-------|-------|--------|--------|---------|--------|--------|----------|
| arXiv | MTask | 13.42 | 3.56 | 11.53 | 0.040 | 0.091 | 0.284 |
| | S2S+LM | 15.25 | 4.62 | 10.41 | 0.052 | 0.273 | 0.399 |
| | SFusion | 16.81 | 5.69 | 10.82 | 0.055 | 0.107 | 0.412 |
| | DialoGPT | 17.84 | 5.20 | 10.68 | **0.296** | **0.711** | 0.208 |
| | StyleDGPT | 19.04 | 5.74 | 12.49 | 0.228 | 0.614 | 0.503 |
| | KASDG | **35.11** | **15.02** | **27.25** | 0.225 | 0.614 | **0.948** |
| Holmes | MTask | 24.47 | 8.87 | 16.03 | 0.027 | 0.063 | 0.276 |
| | S2S+LM | 25.32 | 9.15 | 14.82 | 0.051 | 0.304 | 0.450 |
| | SFusion | 25.91 | 9.68 | 15.87 | 0.045 | 0.098 | 0.479 |
| | DialoGPT | 27.19 | 8.31 | 14.78 | **0.172** | **0.589** | 0.282 |
| | StyleDGPT | 29.58 | 10.15 | 17.10 | 0.101 | 0.452 | **0.602** |
| | KASDG | **36.86** | **13.21** | **22.82** | 0.139 | 0.553 | 0.583 |

Table 1: Automatic evaluation results in the arXiv and Holmes styles.

and arXiv-like and Holmes-like responses (Gao et al., 2019b). TCFC focuses on formality with an informal dialogue corpus and a formal sentence corpus. arXiv and Holmes are constructed by the academic website and Sherlock Holmes novel series, respectively. Only the informal dialogues contain context-response pairs, the other styles are non-parallel corpus of sentences. Both tasks share the same informal Reddit conversation dataset. The detailed statistics are given in Appendix A.1.

### 4.2 Compared Baselines

To verify the effectiveness of the proposed method, we compare our model with the pre-trained dialogue generation model **DialoGPT**, which has not been specifically trained for stylized generation, serving as a baseline for evaluating the performance of models with explicit style transfer capabilities (Zhang et al., 2019). We compare our proposed model with the pipeline method **S2S+BT**, which first generates a non-stylized response using a S2S approach, and then refines the response by applying a text style transfer model (He et al., 2020). Besides, we also compare our proposed model with recent stylized dialogue generation models: **MTask** is a vanilla multi-task learning model (Luan et al., 2017) trained on both $\mathcal{D}_{dia}$ and $\mathcal{D}_{sty}$. **S2S+LM** combines the output distribution of a sequence-to-sequence (S2S) dialogue model with a stylized language model (LM) (Niu and Bansal, 2018). **SFusion** is also a multi-task learning model that bridges the gap between conversation modeling and non-parallel style transfer by sharing a structured latent space (Gao et al., 2019b). **SRJT** is based on the back-translation that incorporates a joint learning strategy and a style routing method to facilitate style transfer in dialogue generation (Zheng et al., 2021). **StyleDGPT** leverages the pre-trained dialogue model and fine-tuned using a combination of

word-level KL loss and sentence-level style classification loss (Yang et al., 2020). **MPDL** explores the interaction between synthetic and original data within a multi-pass dual learning framework (Li et al., 2021). We also discuss the advantages of our model compared with large language model ChatGPT (OpenAI, 2023) in Appendix A.5.

### 4.3 Implementation Details

We implement our experiments using Pytorch with transformers [2] . The main forward model and backward model are initialized with $\text{BART}_{base}$ with the learning rate 5e-05. We use grid search to tune the hyper-parameters according to the performance of validation. The temperature hyperparamter $\tau$ is set to 0.1 and all coefficients $\lambda$ except $\lambda_{s_1}$ are set to 1.0. During training, we leverage the AdamW optimizer with batch size 4. We select $K = 20$ most related style sentences calculated by sentence-BERT [3] as input of style encoder. We warm up the backward model with a freeze step of 5,000 and warmup step of 10,000. Due to the instability of back translation, we use beam search for generation with the beam size 50 and filter out the pseudo context that share more than 30% tokens with its response.

## 5 Experimental Analysis

### 5.1 Main Results

**Automatic Evaluation** Following the previous work, ROUGE and BLEU metric (Papineni et al., 2002) are employed to measure n-gram overlap between the generated responses and the reference responses for automatic evaluation. Distinct (Li et al., 2015) measures the proportion of unique n-grams in the generated responses. To evaluate the style intensity, we follow (Zheng et al., 2021)

---

[2]https://github.com/huggingface/transformers
[3]https://www.sbert.net

| Model | B-1 | B-2 | Dist-2 | BERT | SVM |
|-------|-----|-----|--------|------|-----|
| **Formal-style Response Generation** | | | | | |
| MTask | 6.35 | 0.50 | 29.3 | 37.3 | 50.1 |
| SLM | 12.6 | 0.99 | 42.5 | 85.6 | 87.2 |
| SFusion | 5.51 | 0.28 | **61.0** | 21.9 | 39.0 |
| S2S+BT | 12.1 | 1.25 | 42.0 | 86.3 | 86.8 |
| SRJT | 15.1 | 1.71 | 43.4 | 97.3 | 96.1 |
| MPDL | 16.5 | 2.07 | 51.3 | **98.6** | **97.1** |
| KASDG | **18.4** | **2.85** | 45.7 | 96.9 | 91.6 |
| **Informal-style Response Generation** | | | | | |
| S2S | 6.92 | 0.61 | 54.8 | 70.1 | 60.9 |
| SFusion | 4.61 | 0.22 | 62.8 | 70.3 | 61.1 |
| SRJT | 6.96 | 0.67 | 49.4 | 69.4 | 59.2 |
| MPDL | 7.12 | 0.69 | 49.5 | 70.3 | 60.7 |
| KASDG | **9.48** | **1.66** | **63.5** | **71.0** | **64.7** |

Table 2: Automatic evaluation results in the Formal and Informal styles.

| Model | Fluency | Relevance | Style Cons. |
|-------|---------|-----------|-------------|
| StyleDGPT | 0.63 | 0.55 | 0.61 |
| SRJT | 0.78 | 0.51 | 0.73 |
| KASDG | **0.80** | **0.72** | **0.81** |

Table 3: Human evaluation results in the arXiv style.

and leverage the trained style classifiers BERT and SVM for Informal and Formal styles. The accuracy of the BERT and SVM classifier on the holdout test set are 93.98% and 89.57% respectively for the TCFC. Furthermore, we adhere to the use of the pre-trained discriminative model $p(S|X)$ (Yang et al., 2020) for evaluating the Style Intensity (StyleIn.) of the response in Holmes and arXiv styles.

The automatic results are summarized in the Table 1 and Table 2. Overall, our method achieves the highest BLEU-1 and BLEU-2 scores on the four styles, which shows the superiority of our method. For formal response generation task on the TCFC dataset, compared with the previous state-of-the-art algorithm MPDL, our proposed method improves the BLEU-1 and BLEU-2 scores by 1.9 and 0.78. The reasons are two-fold: (1) KASDG can be better aligned in both style and semantics via the style sentences retrieval process. (2) the feature-guided selection module effectively filters out the noise information. We do not achieve the highest score on Dist, mainly due to the negative correlation between BLEU and Dist. Our methods exhibit superior performance in the arXiv style, while only showing comparable results to StyleDGPT in the Holmes style. This can be attributed to the larger corpus of 1,347k sentences in arXiv compared to 38k in Holmes. The increased availability of style sentences contributes to improved retrieval results and a clearer style signal for our model, resulting in better performance. Training bias leads to better

performance of KASDG in informal style. The supervised learning is applied to informal style, while formal style requires pseudo context generation.

**Human Evaluation** Apart from automatic metrics, we conduct human evaluations to assess the quality of generated stylized responses in arXiv style. Specifically, we randomly sample 100 examples and employed three experts to grade the quality of generated responses and reference responses by three criteria: (1) Fluency measures the generated responses are readable. (2) Relevance measures the response is coherent with the given context. (3) Style Consistency exhibits the desired style in the response. For each human indicator, we give the annotators three ranges: [0,0.33] is very dissatisfied; (0.33,0.67) is general satisfaction; and [0.67,1] is very satisfied. All generated summaries are re-capitalized and de-tokenized fairly.

The results are shown in Table 3, which shows that proposed KASDG outperforms the other baseline models in both human metrics. The kappa statistics are 0.58, 0.53 and 0.55 for Fluency, Relevance and Style-Consistency respectively, indicating the moderate agreement between annotators (Landis and Koch, 1977). Specifically, for the response generation in arXiv style, the style consistency and content relevance are significantly higher than the baseline models, indicating that our feature-guided selection module has gained better modeling ability through displayed guidance.

### 5.2 Ablation Study

To better quantify the effectiveness of each module in KASDG, we conduct an ablation study on the Holmes dataset: From the ablations in Table 4, we observe that: (1) The SRKL and CL both play a very important role in the model and their removal will result in varying degrees of performance degradation; (2) The modified SRKL outperforms the original KL in all metrics, which is a good indication that the SRKL constraint model is optimized towards the target style and achieves better style feature. (3) Contrastive learning enhances our model's dialogue capabilities at the expense of style proficiency, aligning with our objective. We incorporate contrastive learning to render filtered style features more response-oriented. As the contrastive loss contains stylized retrieved hidden as negative samples, our contrastive loss primarily targets content fidelity, ensuring that filtered style features are closely related to the response. How-

| Model | BLEU-1 | BLEU-2 | Rouge-L | Dist-1 | Dist-2 | BERT |
|---|---|---|---|---|---|---|
| Base-model | 35.32 | 11.90 | 22.09 | 0.126 | 0.520 | 89.09 |
| w/ CL | 36.36 | 12.64 | 22.23 | 0.132 | 0.540 | 87.07 |
| w/ KL | 34.88 | 12.33 | 21.56 | 0.136 | 0.548 | 85.76 |
| w/ SRKL | 36.77 | 12.55 | **22.83** | 0.138 | 0.550 | **89.20** |
| w/ CL+SRKL | **36.86** | **13.21** | 22.82 | **0.139** | **0.553** | 86.81 |

Table 4: Ablation Study in the Holmes style. The accuracy of BERT on the holdout test set is 99.4%.

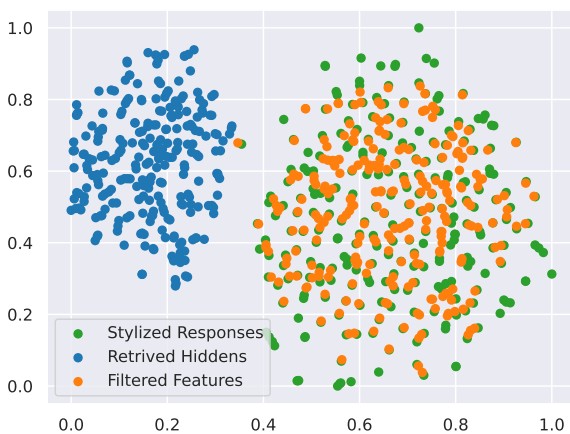

Figure 3: Visualizing the retrieved style hidden, stylized responses and filtered style features in Holmes style.

| Context | I think we need a bit more context. How do you just tackle a teacher and not get in trouble? |
|---|---|
| Human | Well, normally you would have to serve detention, but his serving was administered at the time of the infraction. |
| Retreival | We must go up and have it out with our friend, the professor. |
| MTask | Totally agreed. What class would be your teacher? |
| SFusion | I would be a man of the moment. |
| SDGPT | He was a man of the few days i saw this in the time of no trouble so can you make no remark from this. |
| SRJT | The first difficulty which we had to contend with was the finding of this american's antecedents. |
| KASDG | I would suggest that the professor's conduct should be reported to the police. |

Table 5: Examples of the generated responses by KASDG and other models in the Holmes style.

ever, alterations in the hidden space unavoidably impact style proficiency, explaining the observed decline in style ability upon incorporating the constrastive learning.

## 5.3 The Impact of Feature-Guided Selection Module

Further, we also investigate the impact of feature-guided selection module. The hidden states in Holmes style are visualized, including the retrieved style hidden, the response hidden and the filtered style features. We use the output of the hidden layer as the representation of the sequences and utilize t-SNE algorithm (Maaten and Hinton, 2008) for visualization. As shown in Figure 3, we can see that the retrieved style hidden are relatively far from the target responses and the boundary between filtered style features and stylized responses is not quite clear. These phenomena illustrate the effectiveness of our selection module. There exist noise in the retrieved style hidden, but after the selection module the irrelevant information are filtered out, which in turn better aids generation.

## 5.4 Case Study

Table 5 presents sample responses generated by different models in the Holmes style. Upon comparing the models, we find that MTask produces coherent responses but struggles to capture the desired style and context relevance, likely due to the absence of pre-trained weights. Both SFusion and StyleDGPT generate partially stylized responses but suffer from poor coherence or contextual relevance, possibly due to the instability of the sampling and ranking processes. SRJT effectively generates sentences in the Holmes style but fails to directly address the context, resulting in suboptimal responses. Retrieval represents the retrieved stylized sentence, which includes some stylistic words (e.g., "professor") but lacks a strong contextual connection. In contrast, our model demonstrates superior performance by generating responses that are coherent, contextually relevant, and consistent with the Holmes style. For instance, when discussing the trouble involving teachers, our response refers to "reported to the police", which is highly relevant. Furthermore, the use of "professor" instead of "teacher" along with "conduct" and "reported to the police" are all topical words found in the Sherlock Holmes novel series. Interestingly, the "professor" mentioned in the retrieval process, showcasing the effectiveness of our model.

## 6 Conclusion

In this paper, we propose a knowledge-augmented stylized dialogue generation model with a feature-guided selection module to provide better guidance signals from both semantic and stylistic perspectives. Specifically, we exploit the style knowledge from a style corpus to provide guiding style signal and design a response-related contrastive learning and a style responsiveness Kullback-Leibler loss to enhance the semantic and stylistic features. Experiments on two public benchmark datasets demonstrate that our proposed method has satisfactory results in generating response with desired style in four target styles. In the future, we plan to explore model performance in zero-shot scenarios and to combine the ability of large language models for intelligent dialogue systems.

## Limitations

A main limitation of this work is the availability of data. Although numerous styles exist, the actual usable data is scarce. As with other efforts to generate pseudo-data, we cannot rule out the possibility that pseudo-data generated through back-translation may compromise the authenticity of the generated text. It is therefore important to encourage users to check the relevance of supplementary generated texts. Furthermore, we do not conduct multiple rounds of experiments with stylized dialogue, which is also a very important capability. Due to limited computing resources, we do not set a large batch size or explore more styles. This is very important in the era of large language models. We plan to explore generating more and higher-quality stylized data with the assistance of a large language model, in order to enhance the performance of the dialogue systems. We will examine these issues more comprehensively in the future research.

## Ethics Statement

This paper presents a knowledge-augmented stylized dialogue generation model with the intention to benefit various natural language applications and enhance creative language expression. The datasets used in our study are publicly available and respect privacy standards. In human evaluation, part-time research assistants were fairly compensated, and established evaluation rules were followed. Our approach does not introduce or exacerbate ethical or social biases in generated dialogues. We remain committed to addressing potential ethical concerns and refining our work as needed, adhering to established ethical guidelines and practices within the scientific community.

## Acknowledgements

We would like to thank the anonymous reviewers for their helpful comments and suggestions. This work is supported by National Key R&D Program of China (No. 2021YFC3340303) and National Natural Science Foundation of China (No. 62122089).

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

# A Appendix

## A.1 Datasets Statistics

Table 6 shows the statistics of datasets, including the number of data and the average length in sentence. The average length of context and response is 12.9, 11.3 respectively.

| Style | Informal | Formal | arXiv | Holmes |
|---|---|---|---|---|
| Training | 217K | 500K | 1,347K | 38K |
| Validation | 978 | 978 | 2,000 | 2,000 |
| Test | 978 | 978 | 2,000 | 2,000 |
| Avg.words | 12.9/11.3 | 12.7 | 16.5 | 16.8 |

Table 6: Dataset statistics. "Avg.words" mean the average words length in sentence.

## A.2 Case Study

| Context | Welcome to ultimate disappointment, I'll be your guide! Time to conquer earth! |
|---|---|
| Human | The harmonic convergence is upon us! |
| Retreival | These and other directions await exploration. |
| MTask | I haven't heard of it yet, I haven't heard anything yet. |
| SFusion | Thank you!!! I will be here for you! |
| SRJT | But, the above is not a complete description, only a brief description is required. |
| StyleDGPT | Time to conquer the time travel speed of the time!! |
| Ours | Now we are in a position to conquer all the horizons of our universe. |

Table 7: Case Study in the arXiv style.

The case study of arXiv style is shown in Table 7, entries highlighted in red point to inconsistencies either in style or content, while those in blue represent a harmonious alignment of both. The responses generated by StyleDGPT and SRJT attempt to incorporate the arXiv style, but they lack coherence with the given context and are semantically unclear. For example, StyleDGPT contains the phrase "time travel speed of the time," which is not meaningful in the given context, while SRJT states that "only a brief description is required", which do not address the topic at hand. MTask generates general and plain responses that do not add any relevant information to the conversation. SFusion is informal and does not adhere to the arXiv style. The Retrieval model generates response that contains stylistic elements such as "exploration" and "direction", but they are not contextually relevant to the given conversation. In contrast, our model is both contextually relevant and consistent with the arXiv style. In our model, the words "conquer" and "universe" demonstrate adherence to the given context, while terms like "position" and "horizons" are consistent with the arXiv style.

## A.3 The Impact of Back Translation and Retrieval

In this section, we study the impact of back translation module and different retrieval inputs, as shown in Table 8. The w/o BT do not use back translation, which retrieves the most similar context in the dialogue corpus as its pseudo context for the input style sentence. $\text{FixBT}_C$ and $\text{FixBT}_R$ refers to use the pseudo context or the style sentences to retrieve paired style sentences respectively through a pretrained back translation model. From the results, we find that the back translation model plays an important part in our model due to the lack of parallel data. Besides, comparing $\text{FixBT}_C$ to $\text{FixBT}_R$, we find that it is better to use response to retrieve style sentences from style corpus, it is due to the quality of the pseudo context and back translation. In $\text{FixBT}_R$, the style score is very low and tends to copy phrases in context. This is due to the fixed back translation model leads to all the same pseudo context. In contrast, we jointly train the back translation model with our main model, and with the evolution of the back translation model, it can provide dynamical and more diverse pseudo context, thus making our model more robust and relieve the problem of copying. We also compare our proposed model to retrieval model RAG (Lewis et al., 2020) in Holmes style, which is an unsupervised retrieval-augmented method. As the results demonstrate, our approach substantially outperforms RAG in both style and content metrics due to (1) Our framework takes full advantage of dialogue datasets via our jointly loss, thus producing more coherent sentences. (2) Retrieved sentences are noisy. Our proposed feature-guided selection module tackles noise in retrieved sentences and extracts dialogue-related style signals.

## A.4 The Impact of Attention

To further analyze the selection module, we visualize the attention in the selection module, as shown in Figure 4. For the same context, arXiv and Holmes focus on different words. arXiv focuses on the academic words (stage, process, fulfilled, contract, and proposal), while Holmes focuses more on the daily life words (requested, business, purchase, living, contracted, and job). The former is commonly seen in formal papers, while the latter appears more realistic and conforms to the feature

| Model | BLEU-1 | BLEU-2 | Rouge-L | Dist-1 | Dist-2 | BERT |
|---|---|---|---|---|---|---|
| RAG | 24.55 | 11.35 | 13.88 | 0.306 | 0.759 | 54.64 |
| KASDG | **36.86** | 13.21 | 22.82 | 0.139 | 0.553 | 86.81 |
| w/o BT | 31.43 | 7.98 | 19.62 | 0.037 | 0.108 | 85.35 |
| FixBT$_C$ | 32.62 | 11.00 | 21.67 | 0.154 | 0.570 | **89.62** |
| FixBT$_R$ | 36.23 | **14.95** | **22.95** | **0.178** | **0.651** | 67.30 |

Table 8: The analysis about the Back-translation and Retrieval methods.

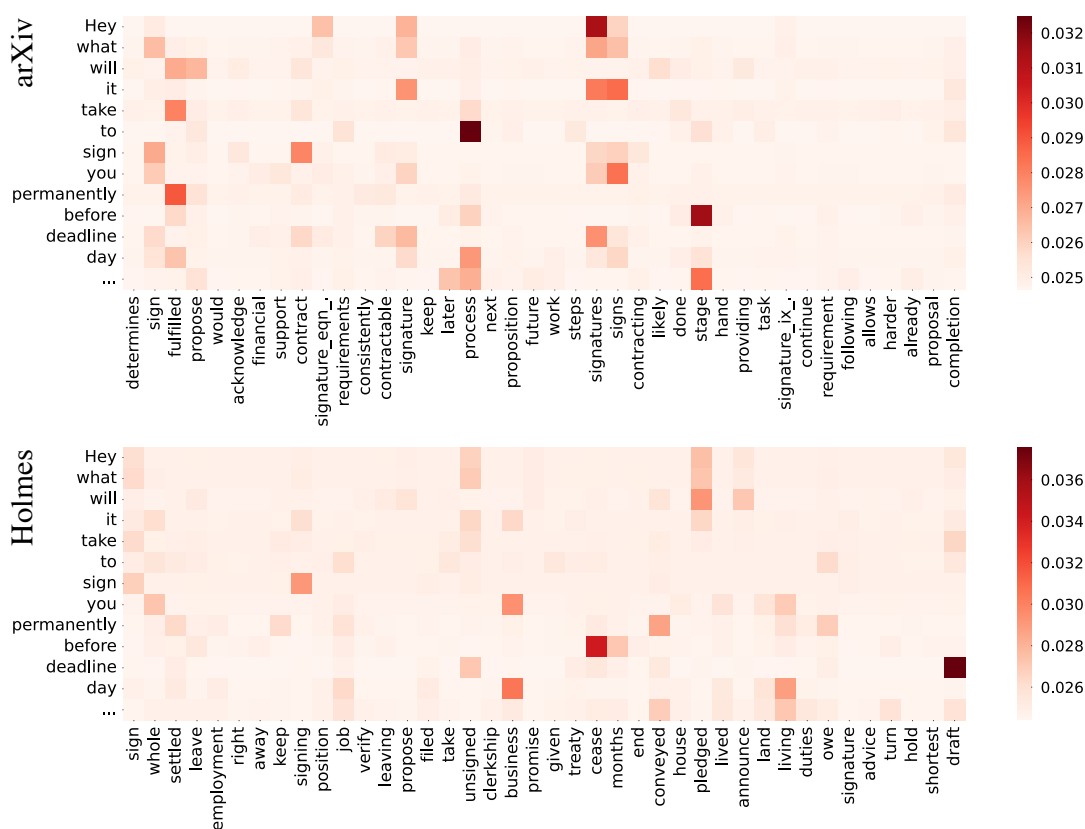

Figure 4: The attention visualization of the selection module for Holmes and arXiv styles.

of Holmes novel. At the same time, we can observe that the words "sign" and "signature" are more activated in arXiv and less activated in Holmes, which is in line with expectations.

## A.5 Analysis of Large Language Model

To further evaluate the performance of our model, we carry out experiments on ChatGPT, a state-of-the-art large language model. Specifically, we design Prompt-1 and Prompt-2 to assess ability of ChatGPT for zero-shot stylized dialogue generation. The results obtained from ChatGPT are summarized in Table 9 (Line 1 - Line 6) and 10. We also take a step forward to evaluate ChatGPT in the setting where examples or explanations of styles are given. For the former, we adopt the classic In-Context Learning algorithm KATE (Liu et al., 2022) with 1,5,10 and 20 examples. For

the latter, we provide two new prompts with style explanations in Holmes style, namely Prompt-3 and Prompt-4, we randomly selected 100 tests in Holmes style to carry out our experiments in advanced setting, as shown in Table9 (Line 7 - Line 13).

*Prompt-1: """You are a stylized conversation agent. Your task is to generate response in style style. The response should be 1-3 sentences. You only need to output the generated stylized response.*
*Context: {context}*
*Response: """*

*Prompt-2: """Generate response in style style. The response should be 1-3 sentences.*
*Context: {context}*
*Response: """*

*Prompt-3: """The Holmes style refers to the deductive and analytical investigative approach pop-*

| Style | Methods | BLEU-1 | BLEU-2 | Rouge-L | Dist-1 | Dist-2 | StyleIn. |
|---|---|---|---|---|---|---|---|
| arXiv | Prompt-1 | 22.09 | 7.71 | 13.27 | 0.186 | 0.506 | 0.092 |
| | Prompt-2 | 23.36 | 8.31 | 13.99 | 0.187 | 0.533 | 0.140 |
| | KASDG | 35.11 | 15.0 | 27.25 | 0.225 | 0.614 | 0.948 |
| Holmes | Prompt-1 | 14.91 | 6.63 | 14.72 | 0.236 | 0.621 | 0.144 |
| | Prompt-2 | 17.38 | 8.10 | 15.54 | 0.272 | 0.662 | 0.086 |
| | KASDG | 36.86 | 13.2 | 22.82 | 0.139 | 0.553 | 0.583 |
| Holmes(100) | Prompt-3 | 22.83 | 5.95 | 12.98 | 0.386 | 0.782 | 0.102 |
| | Prompt-4 | 21.18 | 7.25 | 14.06 | 0.398 | 0.809 | 0.125 |
| | ICL-1shot | 23.27 | 8.00 | 13.44 | 0.337 | 0.714 | 0.141 |
| | ICL-5shot | 25.48 | 9.11 | 13.55 | 0.353 | 0.735 | 0.164 |
| | ICL-10shot | 26.10 | 10.16 | 14.63 | 0.366 | 0.745 | 0.164 |
| | ICL-20shot | 25.21 | 8.92 | 13.24 | 0.376 | 0.747 | 0.250 |
| | KASDG | 36.86 | 13.21 | 22.82 | 0.139 | 0.553 | 0.583 |

Table 9: ChatGPT evaluation results in the Holmes and arXiv styles.

| Methods | BLEU-1 | BLEU-2 | Dist-2 | BERT | SVM |
|---|---|---|---|---|---|
| **Formal-style Response Generation** | | | | | |
| Prompt-1 | 13.7 | 1.37 | 46.7 | 95.9 | 89.7 |
| Prompt-2 | 15.9 | 1.58 | 43.9 | 97.5 | 95.5 |
| KASDG | 18.4 | 2.85 | 45.7 | 96.9 | 91.6 |
| **Informal-style Response Generation** | | | | | |
| Prompt-1 | 6.30 | 0.95 | 40.2 | 26.2 | 30.0 |
| Prompt-2 | 5.37 | 0.83 | 41.4 | 23.9 | 25.2 |
| KASDG | 9.48 | 1.66 | 63.5 | 71.0 | 64.7 |

Table 10: ChatGPT evaluation results in the Formal and Informal styles.

*ularized by the fictional detective Sherlock Holmes, created by Sir Arthur Conan Doyle. This style emphasizes keen observation, logical reasoning, and attention to minute details to solve complex mysteries and crimes. Generate response in Holmes style. The response should be 1-3 sentences.*
*Context: {context}*
*Response: """*

*Prompt-4: """"The Holmes style refers to the distinctive characteristics of the fictional detective Sherlock Holmes, created by Sir Arthur Conan Doyle. Generate response in Holmes style. The response should be 1-3 sentences.*
*Context: {context}*
*Response: """*

The results indicate that our models perform better in terms of both style and semantics, with the exception of formal style where ChatGPT demonstrates a slight advantage. We attribute these outcomes to the following factors: (1) ChatGPT is not specially trained to generate stylized responses, while our model incorporates an optimized style feature selection module that provides clearer style signals. (2) Since ChatGPT has no access to the stylized corpus, it can only retrieve style knowledge from its parameter memory. In contrast, our model performs explicit style knowledge retrieval, fully utilizing the available style corpus to provide

precise style signals. (3) To some extent, the results can be attributed to the distribution of the pretrained corpus. ChatGPT is trained on a diverse range of sentences, enabling it to produce coherent and varied responses. Moreover, ChatGPT has been exposed to a notable amount of formal-style sentences, which explains its superior performance in formal style. However, with fewer instances of other less common styles in its training data, the performance of ChatGPT in those styles is comparatively weaker. It is worth noting that a disparity exists between stylized sentences and the provided style corpus $\mathcal{D}_{sty}$. Consequently, the style knowledge retrieved from its memory is just a statistical approximation of the given style.