# OpenReview forum: "Stylized Dialogue Generation with Feature-Guided Knowledge Augmentation"
_EMNLP/2023/Conference — EMNLP 2023 Findings_

### Official Review · Reviewer_Dqb9 · 2023-08-04

**Soundness:** 4

**Excitement:**

3: Ambivalent: It has merits (e.g., it reports state-of-the-art results, the idea is nice), but there are key weaknesses (e.g., it describes incremental work), and it can significantly benefit from another round of revision. However, I won't object to accepting it if my co-reviewers champion it.

**Paper Topic And Main Contributions:**

This paper try to build a stylized dialogue model using non-stylized dialogue data and stylized non-dialogue texts. The proposed method is built on top of a back-translation process, i.e., the author use a reverse dialogue model to produce possible dialogue contexts to the stylized responses and use these pseudo dialogue pair to train the proposed model.The authors use retrieved text to enhance the stylistic features exhibited in the generated responses. The author also introduce a set of lossed to enhance the hidden space of the retrieved stylized texts. Specificlaly. a  contrastive learning loss is used to enhance the retrieval performance and a KL loss is used to enforce similiar attention distribution between different queries.

**Questions For The Authors:**

1.  what is the metric "StyleIn" in table 1 and table 10?

2. Is the interactive training process used in SRJT used in this method?

3. Why not report the classification score of SVM in table 4? Why not report the classification score of the SVM and BERT classifer in Table1?

**Reasons To Accept:**

1. The investigated problem is interesting.

2. The compare between the proposed model and ChatGPT looks promising.

**Reasons To Reject:**

1. The evaluation of the proposed method is not solid. Specifically, not all baselines are used for all datasets. Some baselines are missing from table 1 and table 2. For example, why not implement SRJT and MPDL for arxiv and holmes style?

2. It is expected to see some analysis of the retrieved stylized sentences. How does these retrieved sentences improve the performance of the generated stylized dialogue responses? is the "Retreival" row in Table5 the retrieved stylized sentence?


**Reproducibility:**

4: Could mostly reproduce the results, but there may be some variation because of sample variance or minor variations in their interpretation of the protocol or method.

**Reviewer Confidence:**

4: Quite sure. I tried to check the important points carefully. It's unlikely, though conceivable, that I missed something that should affect my ratings.

---

> ### Author Rebuttal · Authors · 2023-08-29
>
> Thank you for the valuable comments that help us improve the work. Below we address the concerns mentioned in the review:
>
> > Q1: Not all baselines are used for all datasets.
>
> We appreciate your great effort for completing our experiments. During our literature review, we found that there are mainly two groups of style datasets:
>
> (1) SFusion and StyleDGPT are tested on the Holmes and arXiv datasets, while SRJT and MPDL are tested on the formal and informal datasets. We showcased our state-of-the-art results on both these datasets.
>
> (2) We followed the baseline results from previous papers, where neither SRJT nor MPDL were tested on Holmes and arXiv. We reproduced SRJT across four styles, but its performance on arXiv and Holmes was poor, so we didn't report it in the paper, the results as follows:
>
>  |       | BLEU-1 | BLEU-2 |
> |-------|------|------|
> |Holmes SRJT | 21.98 | 6.15 |
> |Holmes  KASDG| 36.86 | 13.21 |
> |arXiv SRJT | 13.74 | 4.23 |
> |arXiv  KASDG| 35.11 |15.02 |
>
> (3) MPDL requires a style transfer dataset, making it impossible to reproduce on Holmes and arXiv, and furthermore, the code hasn't been made public. It's worth highlighting that our method doesn't require a style transfer dataset, resulting in less data dependency and better generalizability.
>
>
> > Q2: Some analysis of the retrieved stylized sentences. Is the "Retreival" row in Table5 the retrieved stylized sentence?
>
> Yes, the "Retreival" row in Table5 represents the retrieved stylized sentence. We demonstrate an example in Figure 1, which shows how our model integrates stylistic words from different retreived sentences to generate stylized response.
>
> Moreover, we conduct experiments on the contribution of retrieved sentences from the perspective of stylized words coverage. In detail, we evaluated the coverage of words in both references and generated sentences from the Holmes dataset. We first apply RAKE [1] algorithm to extract key stylized words. Then we compute the word coverage of our retrieved top-k sentences (k=1,5,10,20). Additionally, the coverage of important words offered by our context was analyzed. The results are listed as follows:
>
> |       | Top1 | Top5 | Top10 | Top20 | Context |
> |-------|------|------|-------|-------|---------|
> | Reference | 0.69 | 1.75 | 2.64  | 4.03  | 2.58    |
> |Generation | 0.67 | 1.48 | 1.94  | 2.4   | 1.49    |
>
> The table shows that the generation process makes use of the retrieved sentences, thus improving the performance of the generated stylized dialogue responses. We will defintely include this in the revision.
>
> [1] Rose, Stuart, et al. "Automatic keyword extraction from individual documents." Text mining: applications and theory (2010): 1-20.
>
> > Question1: The "StyleIn" in table 1 and table 10?
>
> "StyleIn" stands for Style Intensity. For consistency, we followed the naming convention from Table 2 in StyleDGPT. Style Intensity refers to the score of a style discriminative model based on GPT2.
>
> > Question2: Is the interactive training process used in SRJT used in this method?
>
> Yes, we adopt the interactive training process in our method.
>
> > Question3: The SVM in table 4 and the SVM and BERT classifier in Table1.
>
> Thank you for bringing this to our attention. Table 4 presents our ablation study on Holmes dataset, while prior studies have also refrained from using SVM on Holmes dataset. We find limited style classification capability of SVM compared to BERT in our initial experiments, so we only include the BERT classification results in Table 4.
>
> Table 1 showcases performance of our models and baselines on the arXiv and Holmes datasets. We also followed two prior papers, Sfusion and StyleDGPT to use Style Intensity to calculate the style scores. Furthermore, it is reported in StyleDGPT that style intensity is more accurate due to the capabilities of GPT-2 (Section 5.1 footnote 8), so we only import the Style Intensity score.
>
> In order to reduce these confusions, we will unify the indicators as much as possible in the revision.

---

### Official Review · Reviewer_ZsDS · 2023-08-04

**Typos Grammar Style And Presentation Improvements:** 1. $i$ and $j$ are used inconsistentl…
**Soundness:** 3

**Excitement:**

3: Ambivalent: It has merits (e.g., it reports state-of-the-art results, the idea is nice), but there are key weaknesses (e.g., it describes incremental work), and it can significantly benefit from another round of revision. However, I won't object to accepting it if my co-reviewers champion it.

**Paper Topic And Main Contributions:**

This paper tackles the problem of stylized dialogue generation, where a model needs to generate a response to a query based on a specific style. However, the parallel data is limited, so previous studies have to exploit unpaired style corpus to learn the style implicitly in the latent space or adopt back-translation to build pseudo pairs. Such methods suffer from the noisy and context-agnostic style signals, as the style is not explicitly provided for each response. This paper makes the following contributions:

(1) Introduces an explicit usage of style signals by retrieving similar responses from an external target style corpus.

(2) Proposes a feature-guided selection method to refine the retrieved style feature and filter irrelevant information, including a response-related contrastive learning loss to make the filter style feature close to the response and a Style Responsiveness Kullback-Leibler (SRKL) loss to filter the retrieved style feature based on both the context and the response.

(3) Experiments on two benchmarks with four styles demonstrate that the proposed framework based on BART can generate satisfactory responses with desired styles, showing superior performance over a few baselines.

**Questions For The Authors:**

A. Did you run the experiments with multiple random seeds or test the statistical significance of the improvements in the ablation study?

B. I appreciate the discussion and comparison with ChatGPT. However, I think it may be difficult for ChatGPT to understand the meaning of the target style given only a style word like *Holmes*, leading to a discrepancy between its generation and the downstream task data. Have you tried describing the target style with an explanation or adding one or more in-context examples in the prompt?

**Reasons To Accept:**

1. The proposed idea to retrieve explicit style features from a target style corpus has not been studied for the stylized dialogue generation task, while it has been commonly used in other text generation tasks.
2. The designs regarding filtering out irrelevant information from the retrieved response are intuitive.
3. Experimental results have shown the effectiveness of the proposed model compared to previous work on two benchmarks.

**Reasons To Reject:**

1. My main concern is the actual effects of the sophisticated components of the feature-guided selection module, including a contrastive loss, an SRKL loss composed of a direction loss, and a responsiveness loss. However, the ablation study shows relatively small (sometimes inconsistent across different metrics) improvements the full model made compared with the base model and the ablated variants, and it is unclear whether these improvements are statistically significant. For example, the results of **w/ SRKL** and **w/ CL+SRKL** seem very close.
2. The proposed framework introduced a lot of $\lambda_*$ hyperparameters that balance the effects of different losses. However, how these hyperparameters affect the results and how the values are chosen is unclear. I am also unsure whether the training with the multiple loss components is stable and how well each converges.


**Reproducibility:**

4: Could mostly reproduce the results, but there may be some variation because of sample variance or minor variations in their interpretation of the protocol or method.

**Reviewer Confidence:**

3: Pretty sure, but there's a chance I missed something. Although I have a good feel for this area in general, I did not carefully check the paper's details, e.g., the math, experimental design, or novelty.

---

> ### Author Rebuttal · Authors · 2023-08-29
>
> Thank you for the valuable comments that help us improve the work. We will respond to the questions, and we appreciate it very much if you kindly raise the scores when the concerns are addressed.
>
> > Q1: The actual effects of the feature-guided selection module and the ablated variants.
>
> Thank you for your advice. First, we would like to point out that the feature selection module is effective in practice. Due to the limitations of the automatic evaluation metrics, the score improvement is not very large. However, the output shows superior results in manual evaluation. Taking Holmes dialogue as an example:
>
> *Context: if u were a soup what kind of soup would u be ?*
>
> *w/o CL: what would he like a curry and a cup of coffee ?*
>
> *w/o SRKL: a cup of coffee , a glass of water , a cup of tea , a coffee , and a cigar .*
>
> *KASDG: what would you recommend , mr holmes , a soup ?*
>
> In this case, ommitting SRKL or CL loss causes a deviation from the topic “what kind of soup”. Also, incorporating them will make the response more stylistic, including the phrase “mr holmes” and the way of punctuation.
> Second, our feature selection module is not complex in inference and already fits the posterior probability distribution into context-aware attention.
> Finally, We conducted the statistically significant using t-test:
>
> |                  | w/CL       | w/ SRKL    | w/ CL+SRKL |
> |------------------|------------|------------|------------|
> | P-value   | 0.00465| 0.00014 | 0.00002|
>
> The result shows that these improvements are statistically significant with p-value<0.05. We will include the details in the revision. thank you!
>
> > Q2: The $\lambda^*$ hyperparameters and the loss.
>
> We appreciate your great effort to improve the robustness of our model. In our experiments, based on empirical experience, we directly set all $\lambda^{*}$ in both Formulas (13) and (14) to 1.0, since it achieves good performance already. We also proposed a warmup strategy for $\lambda_{s_{1}}$ in Formula (16) (Line 356). For $\lambda_{s_{0}}$​ and $\lambda_{bt}$​ in Formula (15), we adjusted the parameters through grid search on validation dataset to minimize the influence of hyperparameters. Training with multiple loss components is stable and converges well. For example, our model converges stably at 80k steps on Holmes dataset, with $\mathcal{L_{s_{0}}}$​ and $\mathcal{L_{s_{1}}}$​ converging to 2.7 and 2.8 respectively, and $\mathcal{L_{bt}}$​ converging to 1.5.
>
> > Q3: i and j are used inconsistently in Equation (9) and (11).
>
> Thank you for pointing out! We make the modifications as:
>
> $\mathcal{L_{resp}}=\log\sum\limits_{i=1}^{L_s}exp({h_s^i}^T \cdot h_r)-\log\sum \limits_{i=1}^{L_s}exp({h_s^i}^T \cdot h_c)$
>
> > QuestionA: Multiple random seeds or test the statistical significance.
>
> Yes, we run the experiments on 5 random seeds. Besides, we conducted the statistically significant using t-test and the scores are listed below:
>
> |                  | w/CL       | w/ SRKL    | w/ CL+SRKL |
> |------------------|------------|------------|------------|
> | P-value   | 0.00465| 0.00014 | 0.00002|
>
> The results show that proposed module significantly outperforms baselines with p-value<0.05. We will include the details in the revision. Thank you!
>
> > QuestionB: Target style with an explanation or in-context examples in the prompt.
>
> Thanks for your advice! We added style explanations in the prompt and provided more examples as illustrations, with the detailed listed as follows:
>
> For style explanations, we provide two new prompts:
>
> *"""Prompt1: The Holmes style refers to the deductive and analytical investigative approach popularized by the fictional detective Sherlock Holmes,*
> *created by Sir Arthur Conan Doyle. This style emphasizes keen observation, logical reasoning, and attention to minute details to solve*
> *complex mysteries and crimes.*
>
> *Generate response in Holmes style. The response should be 1-3 sentences.*
>
> *Context: {context}*
>
> *Response: """*
>
> *"""Prompt2: The Holmes style refers to the distinctive characteristics of the fictional detective Sherlock Holmes, created by Sir Arthur Conan Doyle.*
>
> *Generate response in Holmes style. The response should be 1-3 sentences.*
>
> *Context: {context}*
>
> *Response: """*
>
> For In-Context Learning, we adopt the classic in-context learning algorithm KATE [1] with 1,5,10 and 20 shot scenarios. Due to resource and time constraints, we randomly selected 100 tests to carry out our experiments, with results as follows:
>
> |                | BLEU-1 | BLEU-2 | Rouge-L | Dist-1 | Dist-2 | StyleIn. |
> |----------------|--------|--------|---------|--------|--------|---------|
> | Prompt1| 22.83  | 5.95   | 12.98   | 0.386  | 0.782  | 0.102   |
> | Prompt2| 21.18  | 7.25   | 14.06   | 0.398  | 0.809  | 0.125   |
> | ICL 1shot      | 23.27  | 8.00      | 13.44   | 0.337  | 0.714  | 0.141   |
> | ICL 5shot      | 25.48  | 9.11   | 13.55   | 0.353  | 0.735  | 0.164   |
> | ICL 10shot     | 26.10   | 10.16  | 14.63   | 0.366  | 0.745  | 0.164   |
> | ICL 20shot     | 25.21  | 8.92   | 13.24   | 0.376  | 0.747  | 0.250    |
> | KASDG          | 36.86  | 13.21  | 22.82   | 0.139  | 0.553  | 0.583   |
>
> As mentioned in Appendix A.5, our model can achieve better results because: (1) Our Feature-Guided Selection Module provides clearer style signals and (2) Our model is trained on the given style corpus, whereas the distribution of pretrained style information in GPT-4's memory might not completely align with the given corpus. This gap cannot be resolved by simply using explanations or by providing a few style sentences as examples.
>
> [1] Liu, Jiachang, et al. “What Makes Good In-Context Examples for GPT-3?”. ACL2022.

---

### Official Review · Reviewer_ehjU · 2023-08-11

**Soundness:** 3

**Excitement:**

3: Ambivalent: It has merits (e.g., it reports state-of-the-art results, the idea is nice), but there are key weaknesses (e.g., it describes incremental work), and it can significantly benefit from another round of revision. However, I won't object to accepting it if my co-reviewers champion it.

**Missing References:**

The manuscript does not consider the inclusion of the RAG series algorithms, which are prominently recognized in the retrieval-augmented generation domain. Also, the newer KDG work should be considered as well.

Reference:

Lewis, Patrick, et al. "Retrieval-augmented generation for knowledge-intensive nlp tasks." Advances in Neural Information Processing Systems 33 (2020): 9459-9474.

Izacard, Gautier, et al. "Few-shot learning with retrieval augmented language models." arXiv preprint arXiv:2208.03299 (2022).

Qingfeng Sun, Can Xu, Huang Hu, Yujing Wang, Jian Miao, Xiubo Geng, Yining Chen, Fei Xu, and Daxin Jiang. 2022. Stylized Knowledge-Grounded Dialogue Generation via Disentangled Template Rewriting. In Proceedings of the 2022 Conference of the North American Chapter of the Association for Computational Linguistics: Human Language Technologies, pages 3304–3318, Seattle, United States. Association for Computational Linguistics.


**Paper Topic And Main Contributions:**

This paper deals with stylized dialogue generation with feature-guided knowledge augmentation where the external knowledge is retrieved using a joint-tuned retriever (transformer encoder), therefore, during the inference, the content generator would generate the stylized content given the target style sentence.

The main contributions of this paper are:

1. the introduction of the feature-guided selection module (retriever)
2. the way to jointly train both the retriever and content generator via a multi-purpose combined loss.
3. using a back-translation module to augment training data.

**Questions For The Authors:**

A. Could the authors provide the BLEU-3 and BLEU-4 scores for both the proposed method and the baseline approaches?

B. Why the backward model needs to be jointly trained with other modules?

C. How is the "context-aware attention" mechanism integrated within the content generator?

D. What motivated the decision to incorporate the "context-aware attention" in the decoder subsequent to the context encoder?

**Reasons To Accept:**

1. The authors meticulously designed distinct loss functions tailored to specific objectives within the framework.
2. The proposed KASDG framework employs an unsupervised strategy, predominantly through the distance between textual representations, to fine-tune a retriever and generator to enhance generation.
3. The performance of the KASDG appears to surpass the baseline.

**Reasons To Reject:**

1. The baseline methods do not compare with the state-of-the-art in unsupervised retrieval-augmented generation techniques (e.g. RAG (Lewis, Patrick, et al., 2020) or ATLAS (Izacard, Gautier, et al., 2022))
2. The case studies presented in this paper lack compelling evidence, and the colour schemes utilized in the figures require comprehensive elucidation.
3. Incorporating external knowledge may introduce inaccuracies to the results and potentially lead to hallucination issues, a concern not addressed in this paper.
4. The newer KDG methods are not considered in the baseline. e.g. DTR (Sun et al., NAACL 2022)

Reference:

Lewis, Patrick, et al. "Retrieval-augmented generation for knowledge-intensive nlp tasks." Advances in Neural Information Processing Systems 33 (2020): 9459-9474.

Izacard, Gautier, et al. "Few-shot learning with retrieval augmented language models." arXiv preprint arXiv:2208.03299 (2022).

Qingfeng Sun, Can Xu, Huang Hu, Yujing Wang, Jian Miao, Xiubo Geng, Yining Chen, Fei Xu, and Daxin Jiang. 2022. Stylized Knowledge-Grounded Dialogue Generation via Disentangled Template Rewriting. In Proceedings of the 2022 Conference of the North American Chapter of the Association for Computational Linguistics: Human Language Technologies, pages 3304–3318, Seattle, United States. Association for Computational Linguistics.



**Reproducibility:**

4: Could mostly reproduce the results, but there may be some variation because of sample variance or minor variations in their interpretation of the protocol or method.

**Reviewer Confidence:**

4: Quite sure. I tried to check the important points carefully. It's unlikely, though conceivable, that I missed something that should affect my ratings.

---

> ### Author Rebuttal · Authors · 2023-08-29
>
> Thank you for the valuable comments that help us improve the work. We will respond to the questions, and we appreciate it very much if you kindly raise the scores when the concerns are addressed.
>
> > Q1.The baseline methods.
>
> We appreciate your great effort for completing our experiments. It is important to highlight the difference between our methods and the unsupervised methods you mentioned. The latter still need correct stylized responses for supervision, while our method can generate stylized dialogue responses using unpaired style corpus. Thanks for your suggestion, we implement RAG using the pseudo stylized dilogue pairs produced by our back translation model as supervision data. The results are listed as follows:
> |        | BLEU-1 | BLEU-2 | Rouge-L | Dist-1 | Dist-2 | StyleIn. |
> |--------|--------|--------|---------|--------|--------|---------|
> | RAG    | 24.55  | 11.35  | 13.88   | 0.306  | 0.759  | 0.546 |
> | KASDG  | 36.86  | 13.21  | 22.82   | 0.139  | 0.553  | 0.583 |
>
> As the results demonstrate, our approach substantially outperforms RAG in both style and content metrics due to
>
> (1) Our framework takes full advantage of dialogue datasets via our $s_0$ loss, thus producing more coherent sentences.
>
> (2) As stated in Section 3.3 (L240-L243), the retrieved sentences are noisy. Our proposed Feature-Guided Selection Module tackles noise in retrieved sentences and extracts dialogue-related style signals.
>
> We will cite the missing references you mentioned in the revision and supplement these experimental results.
>
> > Q2. The case studies and the colour schemes.
>
> For colour schemes, red indicates inconsistency in style or content, while blue signifies that both are aligned. We present an additional case in Holmes test dataset:
>
> *Context*:
> *aah i did that in a middle school law class, except he ended up crying and we got 10 days of detention.*
>
> *SRJT*:
> *by considering the d brane, we can make the same argument for the bulk.*
>
> *KASDG*:
> *we would like to remind the reader that in this case a student would be in violation of the stephen law.*
>
> We can clearly notice that SRJT's output “same argument” (will be marked red) leads to reduced contextual relevance, while our output maintains style and content alignment well. We will provide more case studies in the forthcoming version of our paper.
>
> > Q3. Incorporating external knowledge may introduce inaccuracies to the results and potentially lead to hallucination issues, a concern not addressed in this paper.
>
> Thanks for pointing out! At present, the hallucination problem is unavoidable in text generation, and what we can do is to increase the relevance of knowledge. To tackle with the hallucination problem, we propose a Feature-Guided Selection Module to filter out information irrelevant to the content, thereby relieving the hallucination problem (L240). As demonstrated by experiments in Table 3, our method achieved the best human scores in terms of Relevance, indicating a reduction in hallucination issues.
>
> > Q4. The newer KDG methods are not considered in the baseline.
>
> Firstly, it's necessary to clarify the differences between DTR’s task and our task. DTR's task is to generate stylized knowledge-enhanced dialogues given a dialogue set, its corresponding knowledge base, and an additional unsupervised style dataset. In contrast, we treat the stylized corpus as knowledge to address the data scarcity problem without utilizing any external knowledge base. Secondly, we still tried to compare DTR but we find that the code link provided in the article is no longer active. We will contact the authors and reproduce the results from this paper in the future, using it as a baseline for comparison.
>
> > QuestionA: The BLEU-3 and BLEU-4 scores.
>
> Thanks for your great advice! We tested the results and reproduced some of the baseline outcomes as follows:
>
> |         | Holmes BLEU-3 | Holmes BLEU-4 |arXiv BLEU-3 | arXiv BLEU-4 |
> |---------|--------|--------|--------|--------|
> | SRJT| 1.46   | 0.00      | 1.17   | 0.00      |
> | SFusion | 1.60    | 0.73   | 1.60    | 0.83   |
> | StyleDGPT| 1.88   | 0.63   | 2.69   | 1.08   |
> | KASDG | 3.39   | 1.48   | 5.72   | 3.25   |
>
> Our methods significantly outperform all baselines on BLEU-3 and BLEU-4 as we provide explicit style knowledge for each response instead of merely a style hidden.
>
> > QuestionB: Why the backward model needs to be jointly trained with other modules?
>
> The reason is that jointly trained backward model, compared to fixed ones, can generate more diverse and dynamic pseudo data. By training on a more varied set of pseudo data, the model becomes more robust and alleviates the issue of copying.
> In Appendix A.3, we conducted an ablation study on the effects of back translation, including whether to involve joint training, and whether we retrieve using context or response during training. As demonstrated in Table 9, joint training generally performs better in terms of content and style.
>
> > QuestionC: How is the "context-aware attention" mechanism integrated within the content generator?
>
> We appreciate the reviewer for raising a question about the integration of the "context-aware attention" mechanism within the content generator. However, we suspect there might be a misunderstanding or typographical error in the question. It seems that the reviewer might be referring to "content-aware attention" instead of "context-aware attention". To clarify, “context-aware attention” is a mechanism we employed in the Feature-Guided Selection Module. Its primary purpose is to filter and extract useful style features under the guidance of various losses, as elucidated in Formula (4).
>
> On the other hand, the "content-aware attention" and "style-aware attention" are integrated within the decoder. Both these attention mechanisms implement the cross-attention structure in transformers. To control the style and context contributions in each layer, we add a gate after the style-aware attention module. The equation of controller gate $CT$ is:
>
> $CT(h_s, h_c) = \lambda \cdot LN(h_s) + (1 - \lambda) \cdot h_c $
>
> $\lambda = \sigma(w \cdot [h_s; h_c])$
>
> Where $h_s$ , $h_c$ denote the output of the style-aware attention and the residual from the previous block. $LN$ represents Layer Norm. $w$ is a learnable parameter and $\sigma$ denotes sigmoid function.
>
> > QuestionD: What motivated the decision to incorporate the "context-aware attention" in the decoder subsequent to the context encoder?
>
> We would like to first kindly point out that it is “content-aware attention” instead of “context-aware attention” in decoder. "Content-aware attention" is essentially the cross attention in the transformer decoder, which provides content information for response generation. While Feature-Guided Selection Module extracts style information useful for the dialogue, it will not cover the full content information in context. Hence, the decoder needs to generate response through a representation of the dialogue history (content-aware attention).

---

### Meta-Review · Area_Chair_6xWz · 2023-09-19

**Recommendation:** 3

**Metareview:**

The detailed and careful design of learning objectives in the proposed KASDG framework tailored to specific objectives is a key contribution of the paper.

As one reviewer pointed out, the paper was missing several important references (RAG, ATLAS, and arguably DTR). While the author provided some comparison (both empirical and theoretical) during rebuttal, these put the thoroughness of the paper's literature review questionable.

---

### Meta-Review · Senior_Area_Chairs · 2023-10-05

**Recommendation:** 3

**Metareview:**

meta review

---

### Decision · Program_Chairs · 2023-10-07

**Decision:**

Accept-Findings

**Comment:**

The detailed and careful design of learning objectives in the proposed KASDG framework tailored to specific objectives is a key contribution of the paper.

As one reviewer pointed out, the paper was missing several important references (RAG, ATLAS, and arguably DTR). While the author provided some comparison (both empirical and theoretical) during rebuttal, these put the thoroughness of the paper's literature review questionable.|meta review